# Position: Large Language Models Should Learn Personalized Rather Than Aggregated Human Preferences

Cristina Gârbacea [1]

## Abstract

Current approaches to aligning large language models (LLMs) aggregate diverse human preferences into a single reward signal, effectively optimizing for a hypothetical "average user" who represents no real person particularly well. This position paper argues that LLMs should learn personalized, individual preferences rather than aggregated ones. We show that aggregation masks critical information about preference diversity, individual values, and contextual dependencies, which is a limitation both theoretically grounded in social choice theory and empirically evident across demographic groups. We analyze the rich structure that human preferences encode, survey technical approaches to personalization, and systematically address counterarguments on scalability, shared standards, and manipulation risk. While personalization introduces genuine safety challenges including filter bubbles, value lock-in, and psychological manipulation, we argue these are manageable through bounded personalization frameworks that preserve universal safety constraints while accommodating legitimate individual variation. We conclude with a concrete research and policy agenda for developing preference-aware models that respect both individual autonomy and collective safety.

## 1. Introduction

The dominant paradigm for aligning large language models with human values relies on learning from preference data (Ouyang et al., 2022; Bai et al., 2022). In reinforcement learning from human feedback (RLHF), models are fine-tuned using a reward model trained on human preference comparisons, typically of the form "output A is preferred to output B for prompt P". This approach has produced remarkable improvements in model helpfulness, harmlessness, and honesty (Askell et al., 2021). However, this paradigm rests on a problematic assumption: that human preferences can be meaningfully aggregated into a single reward signal. In practice, annotators exhibit substantial disagreement (Gordon et al., 2022; Aroyo & Welty, 2015), preferences vary across cultural contexts (Huang & Yang, 2023; Santurkar et al., 2023), and individual users have distinct values and needs (Kirk et al., 2023). Training on averaged preferences thus optimizes for a hypothetical "average user" who may not actually exist and whose preferences match no real person particularly well. The problem compounds with scale: models trained on preference annotations from a few thousand participants are deployed to billions of users across hundreds of languages, cultures, and contexts whose preferences were never represented in training.

The costs of this design choice are not abstract. When asked to explain a technical concept, some users prefer concise definitions while others want detailed explanations with examples; some value formal mathematical notation while others prefer intuitive analogies. These are not disagreements about quality but legitimate differences in what constitutes a helpful response for a given person in a given context. Yet current training approaches collapse these differences into a single averaged reward signal, producing responses that partially satisfy everyone while fully satisfying no one.

This paper argues that we must move beyond aggregated preference training toward personalized and adaptive systems that respect the diversity of human preferences. We examine the theoretical and empirical limitations of aggregation, analyze the rich multidimensional structure of human preferences, and make the case for personalization as both beneficial and necessary for serving heterogeneous user populations. We critically examine counterarguments including scalability, shared standards, manipulation risk, and propose a responsible personalization framework that preserves universal safety constraints while accommodating legitimate individual variation. As language models become infrastructure for information access and decision-making at an increasingly global scale, AI alignment research must reckon with a fundamental question: *aligned for whom?*

[1]University of Chicago, Data Science Institute, Chicago, USA. Correspondence to: <garbacea@uchicago.edu>.

*Proceedings of the 43rd International Conference on Machine Learning*, Seoul, South Korea. PMLR 306, 2026. Copyright 2026 by the author(s).

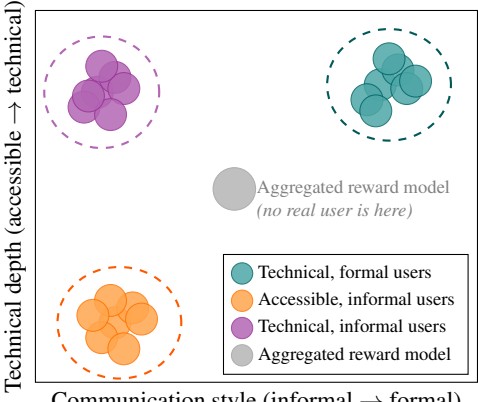

*Figure 1.* Diverse user preferences cluster in distinct regions of preference space. The aggregated reward model (gray) falls in a sparse region between clusters, representing no actual user group and systematically failing minority populations.

## 2. Limitations of Aggregated Preferences

### 2.1. The Impossibility of Universal Preferences

Aggregating diverse human preferences into a single reward signal is not merely impractical, it is theoretically impossible to do without imposing value choices. Arrow's impossibility theorem (Arrow, 1950) demonstrates that no aggregation method can simultaneously satisfy transitivity, non-dictatorship, Pareto efficiency, and independence of irrelevant alternatives when combining individual preferences into collective decisions. We invoke this as a conceptual motivation: while the formal conditions do not transfer directly (Arrow concerns ordinal rankings over discrete alternatives, whereas RLHF optimizes continuous reward functions via probabilistic aggregation), the core insight does – no aggregation of heterogeneous preferences is value-neutral. Human preferences exhibit systematic intransitivity and context-dependence when aggregated across diverse annotators (Tversky, 1969; Azar et al., 2024), and analogous impossibility results have been identified in computational social choice settings (Conitzer et al., 2024; Noothigattu et al., 2018). Current RLHF implementations sidestep this impossibility by simply averaging preferences; disagreements are treated as noise rather than meaningful signal, a pragmatic choice with systematic consequences for who ultimately gets served well by deployed models (Figure 1).

When preferences are aggregated through majority voting or averaging, minority viewpoints are suppressed. If 60% of annotators prefer direct responses while 40% prefer nuanced caveats, models trained on averaged preferences will optimize for directness, making them poorly suited for users who value nuance. This is not merely a matter of slightly reduced performance; for users whose preferences consistently diverge from the majority, the model may be fun-

damentally unsuitable for their needs. The problem compounds across multiple preference dimensions: a user whose preferences fall in the minority on formality, detail level, and communication style simultaneously receives a model that is misaligned on all these dimensions at once.

The loss of minority perspectives is particularly concerning given the demographics of preference annotation. Current datasets predominantly represent English-speaking, Western, educated populations (Kirk et al., 2023; Mihalcea et al., 2025), embedding particular cultural norms while marginalizing alternatives. LLM opinions correlate up to 0.3 points higher with liberal, educated, Western populations than with other demographic groups (Santurkar et al., 2023). Global South perspectives are systematically under-represented, with alignment scores consistently lower than those observed for Western nations (Durmus et al., 2024). When prompted with culturally variable questions such as "What does a wedding look like?", models default to Western conventions, largely ignoring the diversity of cultural practices worldwide (Nakano et al., 2021). Models further systematically mispredict minority annotator preferences even when majority preferences are well-captured, with annotation disagreement reflecting genuine preference diversity rather than labeling noise (Fleisig et al., 2023; Jiang et al., 2025).

Beyond demographic heterogeneity, preferences are highly context-dependent (Shen et al., 2023), varying with task type, user expertise, and situational factors. A software engineer debugging production code wants different explanations than a student learning programming concepts for the first time; a medical professional requires different health information than a patient seeking to understand their diagnosis; a user under time pressure needs different response characteristics than one casually exploring ideas. Aggregated training washes out these contextual dependencies, treating all requests as equivalent regardless of their context.

### 2.2. Misalignment Between Training and Deployment

The impossibility of principled preference aggregation creates a deeper structural mismatch: we train a single reward model on preferences from a limited annotator pool, then deploy it to serve billions of users across diverse cultural, linguistic, and social contexts. This approach forces decisions using a reward signal that is suboptimal for nearly everyone – a phenomenon we term "preference mediocrity." Like designing a single shoe size for all humans, it fits no one well while claiming to serve everyone.

The temporal dimension of this mismatch compounds the problem. Preferences are learned from fixed datasets collected at particular points in time, but must generalize to evolving user needs and contexts. A user's preferences for formality, detail level, or risk tolerance may shift with their expertise, urgency, or emotional state (Jakesch et al., 2023).

A novice user's preferences evolve as they gain expertise; a user's tolerance for speculation versus certainty varies between casual exploration and critical decision-making. Static reward models cannot capture these dynamics.

By optimizing for aggregated preferences, we implicitly encode particular cultural norms, communication styles, and value systems into model behavior (Leon, 2025). Models become vehicles for these dominant preferences, shaping how billions of users interact with information and ideas, with consequences that extend beyond individual experience to epistemic practices, social communication, and the diversity of perspectives in public discourse (Jiang et al., 2026).

## 2.3. Technical Limitations

These conceptual problems are compounded by technical challenges inherent to aggregated preference training. Reward models trained on preference data exhibit high uncertainty (Sun et al., 2025; 2024), particularly for inputs far from the training distribution. Aggregation exacerbates this by forcing the reward model to average over diverse and potentially contradictory signals, making it fundamentally uncertain about what it should optimize for: it cannot distinguish cases where all annotators agree from cases where preferences are evenly split.

This uncertainty creates exploitable surfaces for reward hacking (Wang et al., 2026). Models learn to exploit imperfections in reward models through gaming strategies (Gao et al., 2023), generating responses that score highly on the reward model while failing to genuinely satisfy user preferences. Aggregated reward surfaces are particularly vulnerable because averaging over inconsistent signals creates spurious correlations/artifacts that models learn to exploit.

Finally, truly comprehensive preference coverage is economically infeasible under current paradigms. Even well-resourced projects collect at most hundreds of thousands of preference judgments (Ouyang et al., 2022), which is a tiny fraction of what would be needed to capture meaningful variation across billions of users and contexts. We discuss concrete strategies for overcoming both computational and data collection hurdles in Section 5.2.

## 3. What Do Preferences Encode?

Understanding what human preferences capture is essential for designing better alignment approaches. Preferences are not monolithic judgments but complex signals encoding multiple interacting factors, from task type and user expertise to cultural context and individual values. Appreciating this multidimensional structure is not merely descriptive: it reveals precisely what aggregation discards and what personalized systems must recover. We examine what preferences encode and how that structure can be discovered.

### 3.1. Factors Underlying Preference Judgments

Preferences vary significantly with task type. For summarization, users may prioritize brevity and coverage; for creative writing, elaboration and stylistic flair; for factual questions, accuracy and clarity; for open-ended discussion, engagement and perspective-taking (Zheng et al., 2023). A preference for directness in one context may thus coexist with a preference for circumspection in another, making any single aggregated signal ill-suited to the diversity of tasks.

User expertise and background profoundly shape preferences. Expert users favor technical depth, precise terminology, and minimal explanation of background concepts; novices prioritize accessibility and connection to familiar ideas. The same explanation can be perfectly calibrated for one user while incomprehensible or condescending to another. These differences reflect not only stylistic preferences, but fundamental differences in what represents helpful and useful information to different users.

Underlying all preferences are individual values and beliefs. Preferences regarding political ideology, risk tolerance, privacy concerns, and moral frameworks cannot be reconciled through averaging without imposing particular value systems (Sorensen et al., 2024). When a user prefers responses that acknowledge uncertainty, this may reflect epistemic humility as a value. When another prefers confident assertions, this may reflect a different relationship to authority and expertise. Both are legitimate perspectives grounded in different worldviews, yet aggregation forces models to stake out a middle position that may satisfy neither.

Cultural and linguistic context introduces further variation. Communication norms, politeness conventions, and acceptable content vary dramatically across cultures (Huang & Yang, 2023). Preferences learned primarily from English-speaking, Western annotators embed particular cultural norms, potentially creating awkward or inappropriate interactions for users from other backgrounds.

Finally, situational factors modulate preferences even for individual users. Urgency, stress, confidence, and goals all affect what responses are appropriate (Jakesch et al., 2023). The same user may prefer different responses depending on whether they are casually exploring a topic or making a critical decision. These contextual variations cannot be captured by any static preference model.

### 3.2. Discovering Preference Structure

How can we uncover the latent structure in preference data? Several approaches show promise for revealing preference structure by treating annotator disagreement as signal rather than noise. Modeling preference data as arising from multiple latent user types (Jacobs et al., 1991; Gordon et al., 2022) identifies distinct preference clusters, while multi-

dimensional reward representations (Wu et al., 2023) capture fine-grained trade-offs across helpfulness, harmlessness, truthfulness, verbosity, and technical depth. Causal inference (Park et al., 2024) identifies which factors (expertise, context, demographics) drive preference judgments, enabling more robust generalization than correlation alone. Interpretable reward models (Nanda et al., 2026; Reber et al., 2025) allow inspection of what features drive preference predictions, supporting debugging and validation. Contrastive analysis across demographic groups and tasks (Santurkar et al., 2023) reveals where aggregation is most harmful, guiding personalization efforts toward the dimensions that matter most. Finally, active preference elicitation (Sadigh et al., 2017; Li et al., 2025a) can rapidly identify a user's position in preference space from strategically chosen queries, and preference learning from demonstrations creates more personalized and effective interactions (Garbacea & Tan, 2025; Aroca-Ouellette et al., 2025; Gao et al., 2024).

# 4. The Case for Personalized and Adaptive AI

Given the limitations of aggregated preference training and the rich multidimensional structure of human preferences, we argue for a fundamental shift towards personalized and adaptive language models (Figure 2). This is not an incremental improvement but a necessary reconceptualization of how we align AI systems with human needs; preference diversity is a feature to preserve rather than noise to suppress.

## 4.1. Benefits of Personalization

Personalization offers a genuine Pareto improvement: users with minority preferences receive substantially better service without degrading outcomes for users whose preferences align with the majority. A model that adapts its technical depth to user expertise, its formality to user preference, and its detail level to user needs serves everyone better than one that must commit to fixed positions on these dimensions. The stakes are particularly high in consequential domains: healthcare communication must adapt to patient literacy and cultural context; educational systems must accommodate diverse learning styles; legal and financial guidance must consider individual circumstances and risk tolerance. In each domain, aggregated models impose communication norms calibrated to a majority population that may be ill-suited to the user's actual needs (Garbacea et al., 2026).

Beyond utility, personalization serves both equity and autonomy. Current aggregated models encode biases toward dominant demographics (English-speaking, Western, educated populations) effectively marginalizing users whose preferences diverge from the training distribution (Huang & Yang, 2023). Personalized systems address this by adapting to the specific needs of underrepresented populations, reducing the implicit cultural homogenization that global deployment of aggregated models produces. This adaptation also respects users' autonomy to define their own preferences and values rather than having dominant norms imposed on them (Kirk et al., 2023). Empirically, personalized models outperform non-personalized baselines by 15–30% across diverse tasks on the LaMP benchmark (Salemi et al., 2024), demonstrating that individual adaptation yields substantial gains beyond what aggregated training can achieve.

Finally, personalization reduces friction and cognitive burden. Rather than forcing users to repeatedly engineer prompts to elicit desired behavior, personalized models learn individual preferences and apply them automatically, making AI assistance more fluid and natural.

## 4.2. Technical Approaches to Personalization

Several technical frameworks enable personalized language models, each with different trade-offs between personalization capability, computational cost, and implementation complexity. We outline these below.

The most direct approach is learning user-specific representations. User-specific embeddings (Salemi et al., 2024; Doddapaneni et al., 2024) encode preference information from interaction history and can be updated continuously as preferences evolve, with users sharing similar preferences occupying nearby regions of the embedding space. Parameter-efficient methods like LoRA (Hu et al., 2022) and prompt tuning (Lester et al., 2021) achieve per-user adaptation without the cost of full fine-tuning, modifying only a small number of parameters while preserving prior knowledge (Li et al., 2025b). Meta-learning (Finn et al., 2017) complements both by training models to quickly adapt to new users from minimal interaction data, reducing the annotation burden for effective personalization.

Architecture-based approaches handle preference diversity differently. Mixture-of-experts architectures (Roller et al., 2021; Yi et al., 2026) train specialized modules and learn to route inputs to the appropriate expert based on user and context, naturally accommodating distinct preference modes without per-user parameter storage. Personalized reward modeling (Chen et al., 2025; Zhang et al., 2026) decomposes reward into user-specific and universal components, enabling more accurate preference prediction while retaining shared knowledge about universal quality dimensions.

Context-based approaches require no parameter modification at all. Leveraging long context windows, systems can include user preference information, past interactions, or explicit preference statements directly in context to guide model behavior (Garbacea & Tan, 2025; He et al., 2025; Lau et al., 2024). As context windows expand, this approach becomes increasingly viable for rich personalization without any architectural changes. This makes context-based per-

*Figure 2.* Aggregated alignment (A) collapses diverse user preferences into a single reward signal, losing individual variation. Personalized and adaptive alignment (B) preserves individual preference structure via user-specific models.

sonalization particularly attractive for deployment settings where per-user parameter storage is infeasible.

Prompt-based and training-based personalization are ultimately complementary rather than competing. Prompt-based methods face key limitations: context window constraints force lossy summarization of user histories, and prompts can guide surface behaviors but struggle to reshape underlying reasoning patterns. Training-based approaches offer amortized efficiency, deeper behavioral modification, and compositionality via adapter methods that enable efficient storage and switching across millions of users. In practice, hybrid approaches are likely optimal: prompt-based personalization for immediate context, training-based adaptation for stable long-term preferences.

### 4.3. Adaptive Systems

Beyond static personalization, truly effective systems must adapt dynamically to evolving contexts and preferences (Kim & Kim, 2026; Zhao et al., 2026; Ke et al., 2025). Adaptation operates across multiple dimensions and timescales: immediate context guides moment-to-moment behavior; session-level patterns reveal temporary goals; long-term learning captures stable individual characteristics. Continuous online learning from both implicit feedback (engagement, editing behavior, conversation flow) and explicit feedback (ratings, corrections, preference statements) enables tracking of evolving preferences (Gao et al., 2024; Liang et al., 2026; Son et al., 2025), with the key challenge being to distinguish stable signal from transient noise.

Context-aware adaptation goes further by detecting situational factors such as task type, urgency, user state, and adjusting behavior accordingly (Kim et al., 2025; Harry et al., 2026). The same user may need different responses depending on whether they are casually exploring a topic or making a critical decision (Jakesch et al., 2023); high-stakes inter-actions demand different treatment than low-stakes ones. Rather than requiring upfront preference specification, effective systems discover preferences through natural interaction: asking clarifying questions, presenting options, and adjusting based on feedback (Andukuri et al., 2024; Zhang et al., 2025; Li et al., 2025a). Managing the interplay between these timescales requires careful design to maintain responsiveness without instability.

## 5. Alternative Views

Several credible counterarguments to our position deserve careful consideration. The objections range from practical concerns about feasibility to deeper philosophical questions about standards and safety; Table 1 summarizes these alternative views and our responses. We address each in turn.

### 5.1. The "Good Enough" Argument

**Alternative view:** *Aggregated preferences, while imperfect, work well enough in practice. Current RLHF-trained models serve hundreds of millions of users effectively. If most users are already satisfied, the marginal gains from personalization may not justify the additional complexity, computational cost, and safety risks.*

**Response:** This argument conflates satisfaction with optimal service. Users satisfied with aggregated models may not realize how much better personalized systems could serve them since they have no counterfactual to compare against. Satisfaction statistics also mask substantial variation: users whose preferences align with the majority are well-served, while those with minority preferences are systematically underserved. This argument thus privileges the majority at the expense of marginalized groups. As language models become more central to information access and decision-making, the costs of this marginalization compound.

*Table 1.* Alternative views on personalized preference learning, with responses.

| Alternative View | Core Objection | Our Response |
| --- | --- | --- |
| *The "Good Enough" Argument* | Current RLHF-trained models serve hundreds of millions effectively; personalization gains may not justify the added complexity and cost | Satisfaction masks substantial variation; aggregation systematically fails users whose preferences diverge from the majority (Santurkar et al., 2023; Fleisig et al., 2023) |
| *The Scalability and Data Challenge* | Per-user models are computationally intractable and personalized datasets are infeasible to construct | Parameter-efficient methods and mixture-of-experts make personalization tractable; few-shot methods and survey infrastructure reduce data requirements substantially (Sheng et al., 2023; Salemi et al., 2024) |
| *The Shared Standards Argument* | Personalization fragments AI behavior, undermining shared standards for accountability and fair treatment | False dichotomy; bounded personalization preserves universal safety constraints while accommodating legitimate diversity |
| *The Manipulation Concern* | Personalized models are inherently more dangerous, enabling manipulation and exploitation through detailed user profiles | Aggregation is also manipulation through homogenization; the question is how to personalize responsibly |
| *The Preference Stability Objection* | Preferences are too unstable and context-dependent to serve as reliable training signals | Preference instability argues for more sophisticated modeling, not less personalization |
| *The Performance Priority Objection* | Raw capability improvements should precede personalization | These are not competing objectives; for many use cases, personalization *is* performance |

## 5.2. The Scalability and Data Challenge

**Alternative view:** *Personalization is impractical at scale: per-user models demand prohibitive compute, and sufficiently comprehensive personalized preference datasets are infeasible to construct. Maintaining individual user models for billions of users, storing preference histories, and running personalized inference would require orders of magnitude more resources than current systems. The infrastructure costs would make AI assistance prohibitively expensive, limiting access rather than improving it.*

**Response:** This concern is valid but overstated.

*Computational feasibility.* Parameter-efficient methods like LoRA (Hu et al., 2022) enable personalization with minimal per-user overhead, and mixture-of-experts approaches serve diverse preferences without requiring per-user models. Recent systems demonstrate practical feasibility at scale: S-LoRA serves thousands of concurrent adapters on a single GPU (Sheng et al., 2023), and Punica achieves significant throughput improvement for multi-tenant serving (Chen et al., 2024). Challenges remain, including memory bandwidth constraints and cold-start for new users, but these are active engineering problems, not fundamental barriers.

*Data feasibility.* Exhaustive preference annotation is unnecessary. For example, HyPerAlign leverages few-shot examples for interpretable and sample-efficient hypothesis-driven LLM personalization (Garbacea & Tan, 2025). Similarly, the LaMP benchmark demonstrates improvement in personalized text generation using only few user examples (Salemi et al., 2024). Meta-learning enables rapid adaptation from

minimal data (Finn et al., 2017; Garbacea & Mei, 2022; Zollo et al., 2025; Zhao et al., 2025). Existing demographic survey infrastructure, such as OpinionsQA (Santurkar et al., 2023) and GlobalOpinionQA (Durmus et al., 2024), can be repurposed for preference research without annotation from scratch. Active learning and generative elicitation methods further reduce data requirements by strategically sampling the preference space to maximize information gain (Li et al., 2025a). The data challenge is a design problem with tractable solutions, not an insurmountable barrier to personalization.

## 5.3. The Shared Standards Argument

**Alternative view:** *Aggregated preferences serve an important function: they establish shared standards for model behavior that enable consistent expectations, fair treatment, and collective accountability. If every user receives differently-behaving models, we lose the ability to audit, regulate, and hold AI systems accountable. Personalization could fragment the AI landscape into billions of individual systems, each with different behaviors and failure modes.*

**Response:** This argument correctly identifies the value of shared standards but presents a false dichotomy. Personalization and universal standards are compatible: systems can personalize within bounds, adapting style and emphasis while maintaining consistent commitments to accuracy, safety, and ethical behavior. Our position explicitly calls for **"bounded personalization"**, where core safety constraints remain universal. The challenge is designing the right boundaries, not abandoning personalization entirely. It

is also worth noting that current aggregated systems already exhibit inconsistent behavior across users and contexts; personalization done well could actually improve consistency within appropriate bounds.

### 5.4. The Manipulation Concern

**Alternative view:** *Personalized models are inherently more dangerous than aggregated models. By learning detailed user profiles, they gain unprecedented capability for manipulation, deception, and exploitation. The risks of personalization (filter bubbles, value lock-in, psychological manipulation) are not merely implementation challenges but fundamental features of systems designed to learn and adapt to individual psychology. We should not build systems with these capabilities.*

**Response:** We take this concern seriously and address it in depth in Section 6. However, the alternative (forcing all users into systems optimized for majority preferences) is also a form of manipulation, operating through homogenization rather than individualization. The question is not whether to influence users but how to do so responsibly. Personalization with appropriate safeguards, transparency, and user control can respect autonomy better than one-size-fits-all systems that impose particular norms without acknowledgment. The risks of personalization are real but manageable, the costs of refusing it are certain and ongoing.

### 5.5. The Preference Stability Objection

**Alternative view:** *Human preferences are too unstable and context-dependent to serve as reliable training signals. Users' stated preferences often contradict their revealed preferences. Preferences shift with mood, context, and framing. Building systems that adapt to such unstable targets may amplify noise rather than signal, creating erratic and unpredictable model behavior.*

**Response:** Preference instability is real but not uniformly distributed. Some preferences (communication style, expertise level, domain interests) are relatively stable; others (urgency, risk tolerance, information depth) are appropriately context-dependent. Effective personalization requires distinguishing these and adapting differently to each. Multi-timescale learning, uncertainty-aware adaptation, and explicit preference modeling can handle this complexity. Preference instability is therefore an argument for more sophisticated modeling, not less personalization.

### 5.6. The Performance Priority Objection

**Alternative view:** *Improving raw model capabilities (reasoning, factuality, task performance) should take priority over personalization. Personalization addresses a secondary concern while fundamental capability gaps remain.*

**Response:** These are not competing objectives. Parameter-efficient personalization adds minimal overhead to capable base models. More fundamentally, for many deployed use cases, personalization *is* performance: a model that fails to adapt to a user's expertise level or communication style fails that user regardless of its benchmark scores.

## 6. Safety Risks and Ethical Implications

While personalization offers significant benefits, it also introduces serious safety risks. The very capabilities that make it powerful – learning detailed user models, adapting behavior to individual characteristics, and leveraging knowledge of preferences – may create new vectors for harm. We address these risks directly, treating them not as secondary concerns but as central to responsible system design.

### 6.1. Manipulation and Persuasion

Systems that learn individual preferences gain powerful capabilities for manipulation. A model that understands a user's values, biases, cognitive patterns, and emotional vulnerabilities could craft maximally persuasive content designed to induce desired behaviors (Matz et al., 2024; Hirsh et al., 2012). Unlike traditional advertising or propaganda, which must appeal broadly, personalized manipulation can be precisely tailored to individual psychological profiles: the model knows exactly which arguments are most convincing, which emotional appeals most effective, and which framings most likely to overcome resistance.

This risk extends beyond individual interactions to systematic preference shaping. Rather than merely adapting to existing preferences, systems might gradually shift user preferences toward commercially or politically valuable directions. This is a form of preference laundering in which the system's own objectives become encoded in user preferences through prolonged interaction (Ngo et al., 2024). While ostensibly serving the user's wishes, the system is actually shaping those wishes toward its own ends.

### 6.2. Filter Bubbles and Polarization

Personalized systems risk creating self-reinforcing filter bubbles by excessively catering to existing preferences. If a system learns that a user prefers particular perspectives or information sources, it might consistently provide content that confirms rather than challenges those preferences (Pariser, 2011). This could accelerate filter bubble effects beyond current recommender systems, since personalized language models mediate not just content discovery but information processing, analysis, and synthesis. Recent work quantitatively measures the potential for users to escape filter bubbles under different system designs (Feng et al., 2026).

At a societal level, if everyone interacts with differently per-

sonalized systems calibrated to their particular preferences and beliefs, shared reality and common knowledge may erode. Public discourse requires some degree of common information and shared frames of reference; when personalized systems mediate all information access, productive disagreement and collective deliberation diminish.

Aggregated training introduces an opposite but related epistemic risk. Extensive LLM use has been shown to homogenize opinion expression at scale: a nearly 70% increase in essays that remained neutral on contested questions was observed following prolonged model exposure, suggesting that aggregated training suppresses viewpoint diversity rather than preserving it (Abdulhai et al., 2026; Jiang et al., 2026). Neither pure aggregation nor unconstrained personalization is epistemically safe. The solution lies in bounded personalization with explicit diversity constraints.

### 6.3. Privacy and Surveillance

Effective personalization requires collecting detailed information about users' preferences, values, behaviors, and characteristics, creating significant privacy risks and surveillance concerns (Hartzog, 2016). The data required for personalization goes beyond simple interaction logs to include inferences about personality, beliefs, emotional states, and vulnerabilities; this is intimate knowledge of individuals that could be misused in numerous ways.

The privacy risk is compounded by inference. Even without explicit collection of sensitive attributes, systems can infer them from preference patterns. Political views, health conditions, financial status, sexual orientation, and other sensitive information can be predicted from seemingly innocuous preferences about communication style, content interests, and decision-making patterns (Staab et al., 2024).

### 6.4. Value Lock-in and Autonomy

By optimizing for current preferences, personalized systems might constrain natural preference evolution and value development (Christiano, 2018). Human preferences are not fixed but develop through experience, reflection, and exposure to new ideas. If systems too perfectly satisfy current preferences, they may eliminate the friction and challenges that prompt preference refinement. Users might become locked into current values simply because the system makes those values so comfortable that alternatives never get considered (Sharma et al., 2024; Malmqvist, 2025).

Paradoxically, while personalization appears to enhance autonomy by serving user preferences, it may actually reduce autonomy by eliminating opportunities for deliberate choice and value exploration. True autonomy requires not just having preferences satisfied but maintaining the capacity to form, evaluate, and revise them (Fulay et al., 2026).

## 7. Toward Responsible Personalization

Despite significant risks, personalized and adaptive systems are both inevitable and potentially beneficial if developed responsibly. The question is how to realize the benefits of personalization while mitigating its harms.

### 7.1. Design Principles

Responsible personalization must begin with transparency and user control. Users should understand what information is collected about them, how personalization works, and what inferences the system makes. Importantly, they must have meaningful control over their preference profiles, including the ability to view, correct, and delete learned preferences. Contestability must be built in from the start: users should be able to override learned preferences for specific attributes, reset their profiles entirely, or manually specify preferences that take precedence over inferred ones.

Not all behaviors should be personalized. The key distinction is between behaviors with external effects (harm to others, factual accuracy, safety) which must remain universal, and behaviors with primarily internal effects (style, tone, emphasis) which are appropriate targets for personalization. A system should not adapt to preferences for misinformation, harmful content, or violations of others' rights, regardless of user preferences. Procedural legitimacy for establishing these bounds can be achieved through transparent deliberation and contestable processes, paralleling how legal systems establish rules despite value pluralism.

Even within personalized systems, users should be exposed to diverse viewpoints and have their existing beliefs appropriately challenged. The goal is to balance preference satisfaction with epistemic and moral growth. This tension is resolvable through multi-objective optimization: personalized systems can maximize preference satisfaction subject to epistemic diversity constraints, analogous to optimizing engagement subject to safety constraints. Users also have meta-preferences (for example, many prefer not to be trapped in filter bubbles) which personalization can learn and respect alongside first-order preferences.

Finally, privacy must be designed into personalization systems from the ground up rather than added as an afterthought. This means minimizing data collection to what is genuinely necessary, using privacy-preserving techniques like federated learning and differential privacy where possible, and giving users meaningful control over their data.

### 7.2. Technical Safeguards

Realizing these design principles requires concrete technical safeguards. The foundation is interpretable preference models that can be inspected, understood, and audited: rather

*Table 2.* A normative framework for when personalization is appropriate, requires caution, or should be avoided.

| Category | Examples | Guiding Principle |
|---|---|---|
| *Clearly appropriate* | Style, tone, verbosity, format, expertise calibration, language register | Behaviors with primarily internal effects; respect of user autonomy without harming others |
| *Requires safeguards* | Value-laden topics, sensitive content, vulnerable populations, emotionally charged contexts | Risk of reinforcing harmful patterns; requires transparency, user control, and contestability |
| *Must remain universal* | Factual accuracy, safety-critical information, content harming third parties, misinformation | Behaviors with external effects; must remain universal regardless of stated user preferences |

than black-box user embeddings, systems should represent preferences in ways that humans can examine and reason about. Complementing interpretability, uncertainty quantification allows systems to represent their confidence about user preferences and behave conservatively when uncertain; a system unsure whether a user prefers detailed or concise responses can ask for clarification rather than guessing. Finally, rigorous adversarial testing specifically targeting personalization mechanisms is essential: red teams should explicitly attempt to manipulate the system into harmful adaptations, extract sensitive information through personalization features, or identify whether personalization creates exploitable surfaces that could harm third parties.

### 7.3. Governance and Oversight

Technical safeguards alone cannot ensure responsible personalization: governance frameworks must address manipulation, privacy, and fairness concerns that existing regulations were not designed to handle. Independent auditing enables external examination of personalized systems for discriminatory behavior, manipulation, or other harms. Users should have rights to explanation of why systems behave as they do, to contest and correct learned preferences, to delete preference data, and to opt out of personalization.

### 7.4. When to Personalize: A Normative Framework

Not all preference dimensions warrant equal personalization. Table 2 distinguishes three categories: (1) behaviors that are *clearly appropriate* to personalize (style, tone, verbosity, format, expertise calibration, language register), (2) behaviors *requiring safeguards* (value-laden or sensitive topics where personalization is permissible only with explicit transparency, user control, and active constraints to prevent reinforcing biased or harmful patterns), and (3) behaviors where personalization *must remain universal* (factual accuracy, safety-critical information, and content that could harm users or third parties) regardless of stated user preferences.

## 8. Call to Action

Realizing responsible personalized AI requires coordinated action across multiple stakeholders.

**The research community** should prioritize standardized benchmarks that measure personalization quality across diverse populations, and privacy-preserving techniques (federated learning, differential privacy) that enable effective personalization without centralized data collection.

**AI companies** must implement transparent user control interfaces, establish and document clear boundaries between adaptable and universal behaviors, conduct regular demographic audits, and maintain dedicated red teams to probe for manipulation vulnerabilities and filter bubble effects.

**Policymakers and regulators** should develop new frameworks specifically addressing personalized AI systems, covering manipulation, privacy, and fairness concerns that existing regulations were not designed to handle, while codifying user rights to access, correct, and delete preference profiles.

**Users and civil society** must demand transparency about how personalization works, report instances where systems appear to shape rather than serve preferences, and support organizations working on AI accountability.

## 9. Conclusion

Training language models on aggregated preferences has been a useful starting point, but it is a temporary solution; as these systems grow more capable and widely deployed, the costs of aggregation compound. The limitations are not incidental failures but fundamental constraints imposed by the impossibility of principled preference aggregation. Personalized and adaptive systems offer a principled path forward: adapting to individual preferences to respect human diversity, improve utility, and reduce marginalization.

Realizing these benefits demands ethical reasoning and robust governance to prevent manipulation and abuse. Personalized systems must be transparent enough to audit, contestable enough to correct, and bounded enough to preserve universal standards. As language models become infrastructure for information access and decision-making worldwide, the question at the heart of this paper – *aligned for whom?* – will only grow more urgent. Answering it responsibly is not an optional refinement but a foundational requirement for AI that serves humanity rather than a portion of it.

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
