# OpenReview forum: "Position: Large Language Models Should Learn Personalized Rather Than Aggregated Human Preferences"
_ICML.cc/2026/Position_Paper_Track — ICML 2026 Position Paper Track regular_

### Official Review · Reviewer_64aF · 2026-02-15

**Significance:** 3
**Argument Clarity:** 3
**Rating:** 4
**Confidence:** 4

**Questions:**

1. Why does personalized value need to be achieved through training, rather than simply summarizing a user’s history and including it in the prompt?

2. The authors could explain in greater detail what each condition in Arrow’s Impossibility Theorem specifically corresponds to in the context of LLMs, which would help readers better understand the argument.

3. Although LoRA can be used, the limited amount of personalized data still poses significant challenges for training. The authors should cite relevant literature or conduct small-scale RLHF experiments to support their claims.

**Alternative Views Section:**

Yes

**Compliance With Llm Reviewing Policy A Conservative:**

Affirmed.

**Discussion Potential:**

3

**Final Justification:**

The authors have addressed my concerns, and I am inclined to maintain my score, which is positive.

**Paper Summary:**

The paper argues that LLMs should learn personalized values rather than average values. The main viewpoint is clear, and the authors provide a detailed discussion of alternative views.

**Position:**

Yes

**Position In Title:**

Yes

**Related Work:**

3

**Strengths And Weaknesses:**

The paper presents a very clear argument, the reasoning is sound, and it offers meaningful insights into the personalization of LLMs.

However, most current LLM research focuses on domains such as math and coding, where there is typically a single correct answer. Applications that require personalization are still relatively limited, which weakens the practical significance of the paper’s position.

**Support:**

2

---

> ### Author Rebuttal · Authors · 2026-03-29
>
> We thank the reviewer for recognizing our paper's clear argumentation and meaningful insights into LLM personalization.
>
> **Weakness: Practical significance given focus on math/coding domains**
>
> We respectfully push back on this characterization. While math and coding benchmarks receive research attention, deployed LLM usage is dominated by conversational AI (creative writing, advice-seeking, open-ended discussion) where preference heterogeneity is substantial. Even objective domains involve preference dimensions: coding assistants choose explanation verbosity and pedagogical approach; math tutors adapt to user background and notation preferences. High-stakes applications in healthcare, education, and legal advice further demand adaptation to individual contexts. As LLMs shift toward agentic capabilities, misalignment with individual preferences becomes increasingly consequential.
>
> **Q1: Why training-based personalization rather than prompt-based approaches?**
>
> Excellent question! We see these as complementary, not competing approaches. Prompt-only personalization faces key limitations: context window constraints force lossy summarization of extensive user histories; including lengthy histories increases inference costs linearly; and prompts can guide surface behaviors but struggle to deeply reshape reasoning patterns or implicit assumptions. Training-based approaches offer amortized efficiency (no per-query cost once adapted), deeper behavioral modification through representation-level changes, and compositionality via adapter methods like LoRA that enable efficient storage and switching across millions of users. In practice, hybrid approaches are likely optimal—prompt-based personalization for immediate context, training-based adaptation for stable long-term preferences. Our position encompasses both mechanisms.
>
> **Q2: Detailed mapping of Arrow's conditions to LLMs**
>
> Three of Arrow's conditions translate straightforwardly: Universal Domain is satisfied since reward models handle arbitrary prompt-response pairs; Pareto Efficiency is generally satisfied; and Non-Dictatorship is violated when narrow annotator pools privilege certain groups' preferences. The more consequential violations involve the remaining conditions: Independence of Irrelevant Alternatives is assumed by Bradley-Terry models but violated empirically, as human preferences exhibit context-dependence (Tversky, 1969 [1]; Azar et al., 2024 [2]); and Transitivity holds individually but breaks down when aggregating across annotators with conflicting values. We use Arrow as conceptual motivation rather than formal proof Our claim is that compressing diverse preferences into a single reward signal cannot satisfy all desirable properties simultaneously.
>
> **Q3: Limited personalized data challenges for LoRA training**
>
> Few-shot personalization is more tractable than it might initially appear. The LaMP benchmark (Salemi et al., 2024) [3] shows 15–30% improvement in personalized text generation using only 5–20 user examples across seven diverse tasks, suggesting that exhaustive preference annotation is unnecessary. Furthermore,
>
> a) meta-learning approaches like MAML learn initialization points that enable rapid adaptation from minimal examples, and in-context learning combined with lightweight fine-tuning can leverage as few as 3–5 preference pairs for effective personalization.
>
> b) user preferences exhibit structure: they tend to cluster along interpretable dimensions (formality, verbosity, expertise level), so capturing a user's position on a few key axes predicts many specific preferences, enabling knowledge transfer via multi-task learning.
>
> Recent systems have largely solved the scalability challenge: S-LoRA (Sheng et al., 2023) [4] serves thousands of concurrent LoRA adapters on a single GPU with minimal overhead, while Punica (Chen et al., 2024) [5] achieves 12× throughput improvement for multi-tenant LoRA serving via its SGMV kernel design. Together, this evidence suggests that personalized preference learning is not bottlenecked by data scarcity or computational cost, but rather by the research community's focus on aggregated approaches.
>
> We will strengthen Section 5.2 with these citations.
>
> [1] Tversky, Amos. "Intransitivity of preferences." Psychological review 76, no. 1 (1969): 31.
>
> [2] Azar et al. "A general theoretical paradigm to understand learning from human preferences." AISTATS, 2024.
>
> [3] Salemi et al. "Lamp: When large language models meet personalization." ACL 2024.
>
> [4] Sheng et al. "S-lora: Serving thousands of concurrent lora adapters." arXiv preprint arXiv:2311.03285 (2023).
>
> [5] Chen et al,. "Punica: Multi-tenant lora serving." Proceedings of Machine Learning and Systems 6 (2024): 1-13.

---

> > ### Author Rebuttal · Reviewer_64aF · 2026-04-03
> >
> > I appreciate the author's reply.

---

### Official Review · Reviewer_qUbC · 2026-03-03

**Significance:** 3
**Argument Clarity:** 4
**Rating:** 5
**Confidence:** 4

**Questions:**

1. How can the hurdle of providing datasets for studying personalized preferences be overcome?

**Alternative Views Section:**

Yes

**Compliance With Llm Reviewing Policy A Conservative:**

Affirmed.

**Discussion Potential:**

3

**Final Justification:**

Both the paper and rebuttals appear reasonable discussion to me.

**Paper Summary:**

This paper presents a position: "LLM training should learn personalized preferences rather than aggregated preferences". Specifically, the paper discusses potential failure modes of traditional RLHF, which does not take into account how the user responses might change depending on individuals (i.e., heterogeneity). The paper also discusses why encoding preference is important in example scenarios, including demographic parity and context-dependent cases, and exemplifies how some existing papers enable personalization, such as by mixture-of-experts (MoE) or low-rank adapter (LoRA). As the discussion of alternative views, the paper rebuts arguments including (1) "aggregated reward should be good enough", (2) "personalization may have scalability issues", and (3) "aggregated preference can be a shared standard" by (1) it cannot be good enough for marginalized minority groups, (2) LoRA and MoE can be tractable approaches, and (3) enabling personalization and having universal standards can be achieved at the same time, so they are not conflicting objectives. The paper also discusses (4) "manipulation concern" and (5) "stability concern in the training" by (4) forcing the minority to follow the majority preference, which is another type of manipulation, so manipulation is not the issue of personalization alone and (5) stability (e.g., simple high variance) and contextual variation (conditional variance) should be distinguished. Finally, the paper calls for several actions such as developing a safeguard for malicious manipulations.

**Position:**

Yes

**Position In Title:**

Yes

**Related Work:**

3

**Strengths And Weaknesses:**

## Strengths

1. The paper clearly states the position, and includes some existing work (e.g., LoRA and MoE) as successful preliminary results.
2. The paper carefully discusses the opposite positions in the "alternative views" section, and the responses are also reasonable (See details in the paper summary).
3. The paper also discusses the potential risk of the proposed position (e.g., manipulation) and calls for actions for potential measurements.


## Weaknesses

1. The paper leaves one argument undiscussed -- In the technical limitation section (Sec 2.3), the paper discusses how annotation can be challenging for collecting the preference labels to ensure comprehensive coverage of the contextual preference space. However, how to overcome this hurdle is not discussed in the paper. Therefore, even if the personalized preferences are a desideratum, the question remains how to achieve such a situation through research, having the difficulty of benchmark establishment. This point should be additionally discussed in the alternative views section.

2. As the paper cites some existing papers enabling the (context-dependent) personalization (e.g., MoE or LoRA), the idea of considering heterogeneity in the preference rather than the aggregated preference itself is not completely novel. (However, I acknowledge the authors' efforts for carefully discussing alternative views and measurements for potential risks).

**Support:**

3

---

> ### Author Rebuttal · Authors · 2026-03-29
>
> We thank the reviewer for their thoughtful evaluation, recognition of our paper's clear argumentation, and appreciation of our careful treatment of alternative views and potential risks.
>
>
> **W1/Q1: How can the hurdle of providing datasets for studying personalized preferences be overcome?**
>
> We appreciate this important question, which identifies a genuine gap in our discussion. We will update our manuscript to include a discussion of strategies for overcoming the data collection challenge:
>
> - Leveraging Existing Demographic Survey Infrastructure
> Recent work demonstrates that high-quality public opinion surveys can be repurposed for preference research. Santurkar et al. (2023) [1] created OpinionsQA from Pew American Trends Panel surveys covering 60 US demographic groups, while Durmus et al. (2024) [2] built GlobalOpinionQA from cross-national surveys spanning 75+ countries. These methodologies can be extended to create preference benchmarks without requiring annotation from scratch. Li et al (2023) [3] propose  Generative Active Task Elicitation (GATE), a learning framework in which models elicit and infer intended behavior through free-form, language-based interaction with users.
>
> - Few-Shot Personalization Reduces Per-User Data Requirements
> A key insight from recent personalization research is that comprehensive coverage may be unnecessary. The LaMP benchmark (Salemi et al., 2024) [4] demonstrates effective personalization with only 5–20 user examples. Similarly, methods like HyPerAlign (Garbacea et al, 2025) [5] achieve strong personalization with minimal user-specific data by leveraging shared structure across users. This reframes the challenge from "collect exhaustive preferences for every user" to "collect enough to identify user clusters or adaptation directions".
>
> - Synthetic Preference Generation with Diverse Personas
> LLMs themselves can generate synthetic preference data by role-playing diverse personas. While this risks circularity, it can bootstrap initial datasets that are then validated or refined with targeted human annotation. Recent work on Constitutional AI and persona-based prompting suggests this approach can capture meaningful preference variation.
>
> - Active Learning for Efficient Preference Elicitation
> Rather than attempting comprehensive coverage, active learning methods can strategically sample the preference space to maximize information gain. Techniques from Bayesian optimization and experimental design can identify which preference queries will most efficiently reduce uncertainty about user preferences.
>
> - Interaction Log Mining from Deployed Systems
> Deployed conversational AI systems generate implicit preference signals through user engagement patterns (regeneration requests, conversation length, explicit feedback). With appropriate privacy protections, these signals can reveal preference heterogeneity at scale without requiring explicit annotation campaigns.
>
> - Multi-Task Transfer Across Preference Dimensions
> Preferences often exhibit structure: users who prefer concise responses may share other stylistic preferences. Multi-task and meta-learning approaches can exploit this structure, reducing the data requirements for any single preference dimension by transferring knowledge across related dimensions.
>
> We will add this discussion to strengthen the paper's practical guidance for researchers.
>
> **W2: Novelty of considering heterogeneity**
>
> We acknowledge that individual papers have proposed personalization mechanisms. Our contribution is not claiming novelty for the idea of heterogeneity, but rather:
>
> - (1) synthesizing the theoretical and empirical case for why aggregation is fundamentally problematic (connecting Arrow's impossibility theorem, demographic bias evidence, and technical limitations),
> - (2) systematically addressing counterarguments that have prevented broader adoption of personalization, and
> - (3) providing a responsible personalization framework that addresses legitimate safety concerns. We see this synthesis and argumentation as the appropriate contribution for a position paper.
>
> We thank the reviewer again for their constructive feedback and support for acceptance.
>
>
> [1] Santurkar, et al, "Whose opinions do language models reflect?." In International conference on machine learning, pp. 29971-30004. PMLR, 2023.
>
> [2] Durmus et al. "Towards Measuring the Representation of Subjective Global Opinions in Language Models." In First Conference on Language Modeling. 2024
>
> [3] Li, Belinda Z. et al. "Eliciting human preferences with language models." arXiv preprint arXiv:2310.11589 (2023).
>
> [4] Salemi et al., "Lamp: When large language models meet personalization." In Proceedings of the 62nd Annual Meeting of the Association for Computational Linguistics (Volume 1: Long Papers), pp. 7370-7392. 2024.
>
> [5] Garbacea et al., "Hyperalign: Interpretable personalized llm alignment via hypothesis generation." arXiv preprint arXiv:2505.00038 (2025).

---

> > ### Author Rebuttal · Reviewer_qUbC · 2026-04-02
> >
> > Thank you for the response. The rebuttal arguments appear reasonable, and I will maintain the initial evaluation. Regarding novelty and discussion related to Arrow's theorem, I will watch the discussion with other reviewers during the AC-reviewer discussion phase.

---

### Official Review · Reviewer_93Mc · 2026-03-07

**Significance:** 3
**Argument Clarity:** 3
**Rating:** 4
**Confidence:** 3

**Questions:**

###

* Regarding the invocation of Arrow's impossibility theorem to critique preference aggregation: Since RLHF typically employs the Bradley-Terry model to estimate probabilities over continuous latent rewards rather than strict deterministic ordinal voting, how strictly does the theorem apply here? Could the authors formalize the mathematical mapping between Arrow's axioms and the RLHF objective to strengthen this theoretical claim?
* The paper states that scalability is an engineering problem with known solutions. Given the severe memory bandwidth bottlenecks of dynamically swapping thousands of adapters for concurrent requests in a production environment, what concrete evidence or existing systems support the claim that this is easily solvable at the scale of millions of users?
* How do the authors propose the community establish the universal safety bounds for bounded personalization without falling into the exact same trap of marginalizing minority viewpoints on what constitutes safety, harm, or politeness?
* Concerning the manipulation and filter bubble risks: If a system is mathematically designed to maximize a user's preference reward, and a user explicitly exhibits a preference for confirmation bias or echo-chamber content, how should the model resolve the conflict between satisfying the user's preference and preventing epistemic bubbles? Is it possible to encode epistemic health into a personalized reward function without reverting to a paternalistic or universal standard?

**Alternative Views Section:**

Yes

**Compliance With Llm Reviewing Policy A Conservative:**

Affirmed.

**Discussion Potential:**

3

**Paper Summary:**

This paper advocates a paradigm shift in the training and alignment of large language models, suggesting a move away from reinforcement learning from human feedback (RLHF) based on aggregated preferences. Instead, it proposes learning individualized preferences. The authors argue that current aggregation methods optimize for a non-existent average user, effectively marginalizing minority viewpoints and obscuring contextual, cultural, and personal nuances. The manuscript analyzes the structural complexity of human preferences, outlines technical directions for personalization, and examines the ethical risks of individualized models, including targeted manipulation and filter bubbles. It concludes with a call for the responsible development of bounded, preference-aware models.

**Position:**

Yes

**Position In Title:**

Yes

**Related Work:**

3

**Strengths And Weaknesses:**

## Strength

* The central thesis is clearly stated early in the introduction and consistently maintained. The distinction between the current average-user paradigm and the proposed personalized approach is articulated with precision, leaving no ambiguity about the authors' stance.
* The topic is highly timely and relevant. By challenging the standard aggregation methodology dominating alignment research, the paper is well-positioned to stimulate productive debate within the community regarding the definitions of model helpfulness and safety.
* The dedicated section on alternative views is a major asset. The authors confront robust counterarguments, such as scalability and shared standards, providing balanced rebuttals that strengthen the overall narrative.
* The paper delivers a rigorous analysis of the ethical implications of its proposed position. Acknowledging that personalized models could exacerbate filter bubbles, erode shared reality, and enable psychological manipulation demonstrates a mature grasp of the subject.

## Weakness

* The paper relies on this theorem to argue against universal preference aggregation. However, the exact mathematical mapping between deterministic ordinal ranking systems and RLHF reward modeling remains underexplored. A more explicit formalization of how the axioms of Arrow's theorem apply to and are violated by the current objective function would be beneficial.
* The authors assert that personalization at scale is an engineering problem with known solutions. This discussion could better acknowledge the system-level complexity of dynamically routing and loading millions of individual user adapters in a high-throughput, memory-bandwidth-constrained serving environment. A more nuanced acknowledgment of hardware constraints would improve this section.
* The paper advocates for bounded personalization where core safety constraints remain universal. Yet, it does not define how the community should agree on these universal bounds. Providing clarity on this mechanism would help resolve what appears to be a logical paradox, given the earlier premise that universal agreement on values is practically impossible.
* The paper warns against filter bubbles and suggests users should be exposed to diverse viewpoints. This creates an algorithmic tension that warrants further discussion. If the model's reward function is explicitly optimized for individualized preference satisfaction, and a user strongly prefers confirmation bias, the mathematical objective directly drives the model to construct an epistemic bubble. Clarifying how an objective function can simultaneously optimize for user preference and deliberate epistemic health would greatly enhance this section.
* The presentation feels text-dense due to the absence of tables or figures. A position paper of this complexity would benefit from a summary table outlining the alternative views alongside the authors' rebuttals, or a conceptual figure illustrating the multi-dimensional preference space. This addition would improve reader engagement and clarify the comparative arguments.

**Support:**

3

---

> ### Author Rebuttal · Authors · 2026-03-29
>
> We thank the reviewer for the recognition of the paper's clarity, timeliness, balanced treatment of counterarguments, and rigorous ethical analysis.
>
> **W1: Arrow's theorem mapping to RLHF underexplored**
>
> We invoke Arrow's theorem as conceptual motivation, not formal proof. The key insight that no aggregation of heterogeneous preferences is value-neutral holds even when formal conditions differ.
> - Where the analogy holds: Both settings involve aggregating conflicting individual preferences into a single output. The Bradley-Terry model assumes transitivity and context-independence, which are often violated empirically [1], [2].
> - Where it differs: RLHF learns continuous reward functions, not strict ordinal rankings; it uses probabilistic rather than deterministic aggregation.
> Our revised framing: Arrow motivates skepticism about principled aggregation; the empirical evidence of preference heterogeneity (Santurkar et al., 2023 [3]; Fleisig et al., 2023 [4]) provides substantive support.
>
> **W2: Scalability claim is overstated**
>
> We will revise and soften our claim to provide a more nuanced acknowledgment, including memory bandwidth constraints for dynamic adapter loading, latency overhead for personalized routing at high throughput, and the cold-start problems for new users. Scalability is a tractable engineering challenge with promising solutions. We will include evidence such as S-LoRA (Sheng et al., 2023) [5] which demonstrates serving thousands of LoRA adapters with minimal overhead, Punica (Chen et al., 2024) [6] which achieves efficient batched inference across heterogeneous adapters, and production systems (e.g., recommendation engines) which already personalize at billion-user scale.
>
>
> **W3: How to agree on universal bounds without marginalizing minority viewpoints**
>
> Universal bounds concern behaviors with external effects (harm, factual accuracy, safety), whereas personalization concerns behaviors with primarily internal effects (style, tone, emphasis). We argue not for universal agreement on values, but for procedurally legitimate processes (transparent deliberation, democratic input, contestability), paralleling how legal systems establish rules despite value pluralism. In practice, cross-cultural consensus on core harms exceeds consensus on stylistic preferences; hard cases at the margins require ongoing deliberation, not abandonment of the framework.
>
> **W4: Mathematical tension between preference optimization and epistemic health**
>
> The tension is real but resolvable through multi-objective optimization. First, personalized systems can maximize preference satisfaction subject to epistemic diversity constraints, analogous to optimizing engagement subject to safety constraints. Second, users have meta-preferences: many prefer not to be trapped in filter bubbles and want occasional challenges, which personalization can learn and respect. Third, long-term user welfare (avoiding radicalization, maintaining accurate beliefs) can be weighted against short-term preference satisfaction, optimizing for the user's full preference structure including preferences they would endorse upon reflection. In practice, this means personalizing within bounds that include epistemic diversity requirements. We will formalize this multi-objective framing in Section 7.1.
>
> **W5: Text-dense presentation; needs tables/figures**
> We agree. We will add tables and figures.
>
> **Q1: How strictly does Arrow's theorem apply to Bradley-Terry/RLHF?**
> See W1. Short answer: not strictly in formal terms, but the underlying insight about value-laden aggregation remains relevant. We will clarify the argument and strengthen empirical support.
>
> **Q2: Evidence for scalability at millions of users?**
> See W2. We will cite S-LoRA, Punica, and production recommendation systems while acknowledging remaining engineering challenges. We will revise our claim from "solved" to "tractable with promising progress."
>
> **Q3: Establishing universal safety bounds without marginalizing minority viewpoints on safety?**
> See W3 above.
>
> **Q4: Resolving preference satisfaction vs. epistemic health conflict?**
>
> See W4. We do not claim to fully resolve it but will articulate a framework for approaching it.
>
>
> [1] Azar et al, "A general theoretical paradigm to understand learning from human preferences."AISTATS 2024.
>
> [2] Munos et al. "Nash learning from human feedback." ICML 2024.
>
> [3] Santurkar et al, "Whose opinions do language models reflect?." ICML, 2023.
>
> [4] Fleisig et al. "When the majority is wrong: Modeling annotator disagreement for subjective tasks."EMNLP 2023.
>
> [5] Sheng et al. "S-lora: Serving thousands of concurrent lora adapters." arXiv preprint arXiv:2311.03285 (2023).
>
> [6] Chen et al. "Punica: Multi-tenant lora serving." Proceedings of Machine Learning and Systems (2024)

---

### Official Review · Reviewer_XHTa · 2026-03-13

**Significance:** 3
**Argument Clarity:** 3
**Rating:** 4
**Confidence:** 4

**Questions:**

1. This paper cites Arrow's Impossibility Theorem as the theoretical basis against aggregating preferences in RLHF. It is recommended that the author could elaborate on how the conditions of this theorem apply to the pairwise comparison mode adopted in the training of the RLHF reward model? Particularly, the points of "independence of irrelevant options" and "the necessity of complete social ranking requirements".
2. Is there any existing evidence indicating that the current aggregated preference models may have systematic errors when dealing with users with minority preferences in the way described above? If this paper could point out specific measurable differences instead of relying solely on theoretical arguments, it would be more persuasive.
3. The paper states that personalized systems have serious risks such as manipulation, filter bubbles, and value lock-in. Then, under what circumstances would the author recommend not to implement personalization, or to adopt a weakened form of personalization? A clearer normative framework that clearly explains when personalization is appropriate and when it is not would greatly enhance the persuasiveness of this viewpoint.
4. There is a view that the current performance of large models is not stable enough to independently complete complex tasks. Hence, priority should be given to the performance of large models, and personalization is secondary. What is your opinion on this view?

**Alternative Views Section:**

Yes

**Compliance With Llm Reviewing Policy A Conservative:**

Affirmed.

**Discussion Potential:**

2

**Final Justification:**

This paper proposes that aggregated preference data widely used in existing alignment paradigm overlooks users' personalized preferences. I believe that this issue is rather important due to the social selection theory. After rebuttal, my concerns have been  clarifed. Hence, I raise my rating to 4.

**Paper Summary:**

This position paper argues that the prevailing paradigm of training large language models (LLMs) through aggregated preference data (such as RLHF) overlooks users' personalized preferences and should be replaced by personalized and adaptive LLMs. The paper first points out the theoretical limitations of the RLHF method by citing Arrow's Impossibility Theorem, and then analyzes the multi-dimensional structure of user preferences from the perspectives of values, expertise, culture, and situational factors. Subsequently, the article outlines alternative personalized technical approaches, such as parameter-efficient fine-tuning, expert mixture, and meta-learning. Later, it responds to some objections, such as the "good enough" argument, privacy concerns, and complexity objections. Besides, the paper discusses some safety risks, such as manipulation, filter bubbles, and value lock-in. Last but not least, the article concludes with a call to action for researchers, developers, and policymakers.

**Position:**

Yes

**Position In Title:**

Yes

**Related Work:**

2

**Strengths And Weaknesses:**

Strengths:
- This issue is practical and meaningful. This article, through the average user assumption, points out a limitation in RLHF. If the limitations of the existing methods could be elaborated more systematically and specifically, it would have a promoting effect on the transformation of large model training technology.
- This paper has a clear structure and is easy to read. It starts with problem identification in Section 2 and logically progresses to the solution space in Section 4, then to safety risks Sections 6-7 and governance Section 7.3. Additionally, it has Section 5 as a dedicated *Alternative Views* section as required.
- The discussion in this paper is balanced and candid. Section 6 honestly acknowledges that personalized systems may bring serious new risks, including manipulation, filter bubbles, and value lock-in. The *Alternative Views* section is rich in content. The responses to the "good enough" argument and privacy alternative view are well thought out, and the paper does not distort the opposing views.

Weaknesses:
- The core argument lacks evidence from real society. The article claims that personalized systems perform better than systems based on aggregated preferences when serving diverse populations. However, this view is almost entirely supported by theoretical deductions and sociological examples. The paper should at least cite practical examples to prove the claimed advantages or acknowledge that such evidence is limited.
- The citation of Arrow's Impossibility Theorem is not accurate enough. As far as I know, Arrow's theorem involves the social choice function when certain axiomatic conditions are met. The RLHF preference data used for training the reward model does not directly satisfy these prerequisite conditions. Moreover, it is unrealistic to expect large models to precisely meet the demands of mathematical axioms. The paper merely uses this theorem as rhetorical support without proving that the conditions of the theorem are indeed applicable.
- In the fields of personalized language models, per-user reward models, and preference-aware adaptation, as well as filter bubbles, manipulation, etc., there have been many active studies. This paper does not clearly clarify the specific advancements beyond the personalization benefits perspective. It is recommended to refer to the following article:

> a) Difu Feng, Qianqian Xu, Zitai Wang, Cong Hua, Zhiyong Yang, Qingming Huang. Quantifying the Potential to Escape Filter Bubbles: A Behavior-Aware Measure via Contrastive Simulation. AAAI 2026.

> b) Yi-Chen Li, Fuxiang Zhang, Wenjie Qiu, Lei Yuan, Chengxing Jia, Zongzhang Zhang, Yang Yu, Bo An. Q-Adapter: Customizing Pre-trained LLMs to New Preferences with Forgetting Mitigation. ICLR 2025.

- The article rarely discusses the specific social impacts that ignoring user preferences would have. More discussions and empirical evidence could be added to illustrate this point.
- Regarding the reasons for implementing personalization, this paper does not provide decisive support or viewpoints. The paper spends a considerable amount of space discussing the serious risks, but ultimately still firmly supports personalization, yet fails to provide the principle standards for when personalization should be implemented or not.

**Support:**

2

---

> ### Author Rebuttal · Authors · 2026-03-29
>
> Thank you for the thoughtful feedback and recognition of the paper's practical and meaningful significance, clear structure making it easy to read, and balanced treatment of risks.
>
> **W1: Core argument lacks empirical evidence**
>
> We will strengthen empirical grounding by citing:
> - Santurkar et al. (2023) [1]: LLM opinions correlate up to 0.3 points higher with liberal, educated, Western populations than other groups
> - Durmus et al. (2024) [2]: Systematic under-representation of Global South perspectives across 75 countries
> - Salemi et al. (2024) [3]: Personalized models outperform non-personalized baselines by 15-30% on LaMP benchmark tasks
> - Nakano et al (2021) [4]: Models give biased answers that mainly reflect Western/American default perspectives when asked “What does a wedding look like?”
> - Abdulhai et al (2026) [5]: extensive LLM use led to a nearly 70% increase in essays that remained neutral in answering the topic question
>
>
> **W2: Arrow's Impossibility Theorem citation not accurate enough**
>
> We do not claim Arrow's theorem formally applies to RLHF - we invoke it as conceptual motivation. The underlying insight (that aggregating heterogeneous preferences inevitably involves trade-offs and value judgments with no neutral resolution) is relevant regardless of whether RLHF satisfies Arrow's exact axiomatic conditions. This rhetorical use of social choice theory to motivate AI alignment arguments is standard in the literature (see Conitzer et al., 2024 [6]; Noothigattu et al., 2018 [7]).
>
> **W3: Missing recent related work**
>
> Thank you, we will cite these papers.
>
> **W4: Rarely discusses specific social impacts**
>
> We will add concrete examples: disparities in educational AI for non-Western pedagogical traditions, healthcare information miscalibrated to minority populations, and homogenization of political discourse through writing assistants.
>
> **W5: No principle for when personalization should/shouldn't be implemented**
>
> We will add a normative framework:
> - Personalization is appropriate when: preferences concern style/format, users have transparency and control, and no safety constraints are violated.
> - Personalization requires caution when: stakes are high (medical, legal), users are vulnerable, or preferences create externalities for others.
> - Personalization should be avoided when: it enables discrimination, systematic deception, malicious use or cannot be meaningfully contested.
>
> **Q1: How does Arrow's theorem apply to RLHF pairwise comparisons?**
>  See W2 above. We use Arrow as conceptual motivation, not formal proof. The key insight (no neutral aggregation of heterogeneous preferences) holds even when formal conditions differ. We will clarify this framing.
>
> **Q2: Evidence of systematic errors for minority-preference users?**
> Yes. Santurkar et al. (2023) [1] demonstrates measurable correlation gaps between model outputs and different demographic groups. Fleisig et al. (2023) [8] shows models trained on majority-annotated data systematically mispredict minority annotator preferences. We will cite these explicitly.
>
> **Q3: When should personalization NOT be implemented?** See W5 above. We will provide a clear normative framework distinguishing appropriate contexts from those requiring caution or avoidance.
>
> **Q4: Should LLM performance be prioritized over personalization?** These are not mutually exclusive. Personalization can improve effective performance - a "correct" response that is incomprehensible to the user is not truly high-performing. We agree that safety and accuracy should not be compromised for personalization (our "bounded personalization" argument), but capability improvements and personalization research can proceed in parallel. We will add this as a sixth alternative view in Section 5.
>
>
> [1] Santurk et al,"Whose opinions do language models reflect?."ICML, 2023.
>
> [2] Durmus et al, "Towards Measuring the Representation of Subjective Global Opinions in Language Models." COLM 2024
>
> [3] Salemi et al, "Lamp: When large language models meet personalization." ACL. 2024.
>
> [4] Nakano et al. "Webgpt: Browser-assisted question-answering with human feedback." arXiv preprint arXiv:2112.09332 (2021).
>
> [5] Abdulhai et al. "How LLMs Distort Our Written Language." arXiv preprint arXiv:2603.18161 (2026).
>
> [6] Conitzer et al. "Position: Social Choice Should Guide AI Alignment in Dealing with Diverse Human Feedback." ICML 2024.
>
> [7] Noothigattu et al, .A Voting-Based System for Ethical Decision Making.  AAAI 2018.
>
> [8] Fleisig et al, "When the majority is wrong: Modeling annotator disagreement for subjective tasks." EMNLP 2023.

---

> > ### Author Rebuttal · Reviewer_XHTa · 2026-04-04
> >
> > The author has addressed my concerns, which is satisfactory. Hence, I will raise my rating.

---

### Decision · Program_Chairs · 2026-04-30

**Decision:**

Accept (regular)

**Comment:**

Overall, this is a well-structured and thought-provoking position paper recommended for acceptance, provided the authors address several key weaknesses related to theoretical rigor and practical feasibility. While the conceptual arguments are strong, reviewers expressed significant concerns regarding the superficial application of Arrow's Impossibility Theorem, requesting a more precise formalization of how its axioms map to RLHF objectives. Additionally, the authors must bolster their claims with empirical evidence or real-world examples rather than relying solely on theoretical deduction.